# Use of motorised transport and pathways to childbirth care in health facilities: Evidence from the 2018 Nigeria Demographic and Health Survey

Cephas Ke-on Avoka [1,2]*, Aduragbemi Banke-Thomas[3], Lenka Beňová[4], Emma Radovich [2], Oona M. R. Campbell[2]

1 Faculty of Public Health, Ghana College of Physicians and Surgeons, Accra, Ghana, 2 Faculty of Epidemiology and Population Health, London School of Hygiene and Tropical Medicine, London, United Kingdom, 3 School of Human Sciences, University of Greenwich, London, United Kingdom, 4 Department of Public Health, Institute of Tropical Medicine, Antwerp, Belgium

* avokakeon@gmail.com

## Abstract

In Nigeria, 59% of pregnant women deliver at home, despite evidence about the benefits of childbirth in health facilities. While different modes of transport can be used to access childbirth care, motorised transport guarantees quicker transfer compared to non-motorised forms. Our study uses the 2018 Nigeria Demographic and Health Survey (NDHS) to describe the pathways to childbirth care and the determinants of using motorised transport to reach this care. The most recent live birth of women 15–49 years within the five years preceding the NDHS were included. The main outcome of the study was the use of motorised transport to childbirth. Explanatory variables were women's socio-demographic characteristics and pregnancy-related factors. Descriptive, crude, and adjusted logistic regression analyses were conducted to assess the determinants of use of motorised transport. Overall, 31% of all women in Nigeria used motorised transport to get to their place of childbirth. Among women who delivered in health facilities, 77% used motorised transport; among women referred during childbirth from one facility to another, this was 98%. Among all women, adjusted odds of using motorised transport increased with increasing wealth quintile and educational level. Among women who gave birth in a health facility, there was no difference in the adjusted odds of motorised transport across wealth quintiles or educational status, but higher for women who were referred between health facilities (aOR = 8.87, 95% CI 1.90–41.40). Women who experienced at least one complication of labour/childbirth had higher odds of motorised transport use (aOR = 3.01, 95% CI 2.55–3.55, all women sample). Our study shows that women with higher education and wealth and women travelling to health facilities because of pregnancy complications were more likely to use motorised transport. Obstetric transport interventions targeting particularly vulnerable, less educated, and less privileged pregnant women should bridge the equity gap in accessing childbirth services.

**Data Availability Statement:** The 2018 NDHS dataset was obtained and used with permission

from the DHS programme (https://dhsprogram.com/).

**Funding:** The work done by OMRC was supported by MOMENTUM Safe Surgery in Family Planning and Obstetrics. The work of LB was funded in part by the Research Foundation – Flanders (Fonds Wetenschappelijk Onderzoek) as part of her Senior Postdoctoral Fellowship (award number 1234820N). The funders had no role in study design, data collection and analysis, decision to publish, or preparation of the manuscript.

**Competing interests:** The authors have declared that no competing interests exist.

## Introduction

Nigeria accounts for over 20% of the 800 daily maternal deaths worldwide [1, 2]. Despite strong evidence of the value of giving birth in health facilities, 59% of women in Nigeria still deliver at home [3, 4]. Some pregnant women experience difficulty getting to health facilities, with evidence showing that health facilities of more than 30 minutes travel distance are associated with decreased skilled delivery at birth [5–7]. Delays in reaching an appropriately staffed and equipped health facility increase the risk of maternal mortality [8]. Research from Nigeria also shows long travel to care is associated with maternal mortality and stillbirths [9, 10].

According to the World Health Organization (WHO), all women need first-level maternity care and the option of referral to higher-level facilities when complications arise [11]. For women who may require care at higher-level facilities and are eventually referred, the travel distance and time to reach facilities where they receive the required care is doubled [12]. Lack of transport is a well-documented barrier to facility-based childbirth in sub-Saharan Africa [13]. In addition, a lack of readily available transport to health facilities for pregnant women especially in emergencies, significantly increases the risk of maternal mortality [14]. Campbell et al. proposed a framework of pathways linking women to intrapartum care, highlighting the need for women to give birth in facilities with well-functioning maternal care and effective strategies for linkage to emergency obstetric services [15].

Across low- and middle-income countries (LMICs), common means of obstetric transport include motorcycle ambulances, four-wheel-drive vehicles, formal motor-vehicle ambulances, and bicycles [12, 16]. However, in parts of Nigeria, personal cars and taxis have been reported as the commonest means of transport for women who are referred for childbirth [12]. Compared to motorised options, non-motorised transport options do not transport pregnant women to health facilities in good time [17]. As per available evidence, motorcycle ambulances, for instance, have been reported to reduce referral delays in rural areas by up to 4.5 hours [18]. Also, transport of pregnant women to health facilities for childbirth is associated with increased access to skilled delivery and a reduction in adverse pregnancy outcomes, particularly for women who use motorised transport [16, 19].

Despite the recognition of critical research needs to better understand how women access essential care at birth when needed, referral and transport systems remain poorly documented and poorly researched [20, 21]. Thus far, there is limited availability of large-scale data, such as the periodically reported Demographic and Health Surveys in LMICs, to understand pathways and mode of transport to care. Where these data exist, there are large gaps in key components of referral, mainly because this information is not captured routinely [22]. Understanding travel to reach health facilities and factors that influence transport and referral mechanisms is a critical step needed to develop an effective obstetric transport and referral system [12]. To improve data collection and better understand pathways and modes of transport to care, the standard Woman's Questionnaire of the 2018 Nigeria Demographic and Health Survey (NDHS) was adapted for country priorities [4]. This included a set of questions on obstetric referral and transport developed by co-authors (ER, OMRC, LB) in collaboration with the Federal Ministry of Health of Nigeria and other implementing agencies [4]. Our study aims to describe transport pathways used by all women who give birth in health facilities in Nigeria (including pathways for women who are referred) and to assess the determinants of the use of motorised transport to health facilities for childbirth care using the 2018 NDHS.

## Materials and methods

### Ethics statement

The 2018 NDHS was approved by the National Health Research and Ethics Committee of Nigeria and the Inner-City Fund (ICF) Institutional Review Board. Permission to use the 2018

NDHS data was granted by the DHS programme, and ethical approval for conducting this secondary analysis was provided by the London School of Hygiene and Tropical Medicine (LSHTM MSc Ethics Ref: 22147).

## Data sources

The NDHS is a cross-sectional, nationally representative household survey. The 2018 NDHS implemented by the National Population Commission collected data between August and December 2018 from 40,427 households [4]. The NDHS uses a multilevel, cluster sampling survey design, and individual sampling weights, where needed, to account for this and for differences in response rates to ensure results are representative at the national level. The 2018 NDHS used computer-assisted personal interviewing (CAPI) for data collection by highly trained enumerators. Responses to the woman's questionnaire, as captured in the births recode dataset, were used in this analysis.

## Inclusion criteria, variables, and measurement

Information on all live births within the five years preceding the NDHS, born to all women aged 15–49 years at the time of the survey, were collected. The unit of analysis was restricted to the most recent live birth per woman within the previous five years. Women who gave birth to multiples (twins and triplets) were considered as a single birth for this analysis. S1 Table describes the set of questions included in the adapted women's questionnaire for the 2018 NDHS. Outcome variables of interest for this analysis included place of birth, referral, and transport to the place of birth. Women who came to the final health facility of childbirth from another health facility (i.e., who were moved from one health facility to another, as captured by Q430A response 1 in S1 Table) were considered as having been referred [23].

The main outcome variable was a binary measure of the use of motorised transport to place of childbirth. Women were asked to report the means of transportation used to get to the health facility where they gave birth; those who used motorised transport (private car or truck, taxi/paid driver, motorcycle, tricycle, scooter, boat with motor, ambulance, public transport/bus) were given the value of "1", while those who used non-motorised transport (bicycle, animal-drawn cart, a boat without a motor, carried, walking, other) were given a value of "0". S2 Table describes the distribution of motorised transport options used by women to health facilities for childbirth. To enable us to determine the use of motorised transport in the total sample of respondents in the survey, women who gave birth at home or in other non-facility locations were added to those who used non-motorised transport but were excluded when only women who gave birth in a health facility were assessed.

Explanatory variables included women's socio-demographic characteristics (age at index birth, highest education level, religion, household wealth quintile, marital status at time of survey, place of residence, and region of residence) and pregnancy-related factors at the time of index pregnancy (parity, number of antenatal care visits, and location of antenatal care). We created a binary variable where women who were deemed most likely to have developed at least one complication during labour or childbirth (consisting of women who had a multiple birth, who gave birth via caesarean section, and/or had an early neonatal death [<8 days]) were assigned a value of "1", while women without any of these outcomes were assigned a value of "0". We created a variable for the initial health facility accessed for childbirth by combining the facility of birth of women who came from home and the facility that initiated referral for women who were referred. Other variables included the final facility of childbirth for all referred women.

## Data analysis

The analysis was done using STATA version 17 (StataCorp, College Station, Texas, USA). The *svyset* command was used to account for sampling weights, clustering, and stratification. We hypothesised that the characteristics, patterns, and determinants of use of transport to childbirth would differ between all women in Nigeria and only women who deliver in a health facility, and thus conducted our analysis comparing these two categories of women. We used descriptive statistics, including frequencies and percentages, to summarize the use of motorised transport by socio-demographic and pregnancy-related characteristics, among all women, among those that gave birth in a health facility, and among those who were referred to another health facility in the intrapartum period. We conducted crude and a stepwise adjusted logistic regression analysis to examine the determinants of use of motorised transport among all women and among women who gave birth in a health facility, with associated odds ratios, confidence intervals and p-values.

## Missing values

Among women who delivered in a health facility, 88 women (1% of all facility deliveries) were missing information on where they came from and the means of transport to facility of birth. We excluded these records in our facility-level, bivariate and logistic regression analysis.

## Inclusivity in global research

Additional information regarding the ethical, cultural, and scientific considerations specific to inclusivity in global research is included in the S1 Text.

## Results

### Description of sample

In the 2018 Nigeria DHS, 21,911 women aged 15 to 49 had at least one live birth in the five years preceding the survey. For the most recent live birth, 59% gave birth at home or in a non-facility location, and 9,015 (41%) gave birth in a health facility. Of these 9,015 women, nearly a third each gave birth in government hospitals (33%), government health centres (32%) and private facilities (32%); 2% gave birth in government health posts. Nearly all women (98%) who gave birth in a health facility reported having travelled there from home or some other non-facility location, while 168 women (2%) moved from one health facility to another in the intrapartum period (referred women). Overall, 6,849 women (31% of all women) used some form of motorised transport to go to their place of childbirth. Among women who gave birth in a health facility, 76% of those who came from home and 98% of those referred used motorised transport (Table 1).

### Pathways to health facility-based childbirth

The Sankey diagram in Fig 1 describes the pathways to childbirth care for all women whose most recent birth was in a health facility. Women initially travelled from home to either government hospitals, health centres, health posts or private facilities (left-hand side of Sankey diagram). Varying proportions of women used motorised transport to facilities, highest among women who initially went to a government hospital (90%). Most of the women who were subsequently referred, delivered in either a government hospital or private facility (right-hand side of Sankey diagram).

**Table 1. Sociodemographic, obstetric factors and transport pathways of all women's most recent live birth in the five-year recall period of the 2018 Nigeria DHS.**

| Characteristics | Women's most recent live birth <5 years before the 2018 NDHS (n = 21,911) | | | p-value | Women who gave birth in a health facility (n = 9,015) | | p-value |
|---|---|---|---|---|---|---|---|
| | Total | Non-facility births | Facility births | | Came from home/non-facility location | Came from another health facility | |
| | 21,911 | 12,896 (59%) | 9,015 (41%) | | 8,759 (98.1%)[#] | 168 (1.9%) | |
| | n (column %) | n (column %) | n (column %) | | n (column %) | n (column %) | |
| **Socio-demographic factors** | | | | | | | |
| **Mother's age at birth** | | | | <0.001 | | | 0.996 |
| Less than 20 | 2,672 (12.2) | 1,886 (14.6) | 787 (8.7) | | 765 (8.7) | 15 (8.9) | |
| 20–29 | 10,794 (49.3) | 6,329 (49.1) | 4,465 (49.5) | | 4,343 (49.6) | 82 (48.6) | |
| 30–39 | 7,207 (32.9) | 3,885 (30.1) | 3,322 (36.9) | | 3,228 (36.8) | 63 (37.6) | |
| 40–49 | 1,237 (5.7) | 796 (6.2) | 441 (4.9) | | 424 (4.8) | 8 (4.9) | |
| **Highest education attained** | | | | <0.001 | | | 0.331 |
| No education | 9,738 (44.4) | 8,264 (64.1) | 1,474 (16.4) | | 1,432 (16.4) | 35 (21.0) | |
| Primary education | 3,293 (15.0) | 1,925 (14.9) | 1,367 (15.2) | | 1,335 (15.2) | 21 (12.5) | |
| Secondary or higher | 8,880 (40.5) | 2,707 (21.0) | 6,174 (68.5) | | 5,992 (68.4) | 112 (66.5) | |
| **Religion** | | | | <0.001 | | | 0.293 |
| Christian | 8,344 (38.1) | 2,857 (22.2) | 5,486 (60.9) | | 5,309 (60.6) | 112 (66.5) | |
| Islam | 13,450 (61.4) | 9,955 (77.2) | 3,495 (38.8) | | 3,417 (39.0) | 56 (33.1) | |
| Traditional/Other | 117 (0.5) | 83 (0.6) | 34 (0.4) | | 33 (0.4) | 1 (0.5) | |
| **Wealth index** | | | | <0.001 | | | 0.914 |
| Lowest | 4,716 (21.5) | 4,168 (32.3) | 548 (6.1) | | 534 (6.1) | 12 (7.3) | |
| Second | 4,850 (22.1) | 3,741 (29.0) | 1,109 (12.3) | | 1,081 (12.3) | 22 (13.2) | |
| Middle | 4,448 (20.3) | 2,592 (20.1) | 1,856 (20.6) | | 1,812 (20.7) | 30 (17.9) | |
| Fourth | 4,103 (18.7) | 1,608 (12.5) | 2,495 (27.7) | | 2,419 (27.6) | 49 (29.0) | |
| Highest | 3,794 (17.3) | 787 (6.1) | 3,007 (33.4) | | 2,912 (33.3) | 55 (32.6) | |
| **Marital status** | | | | <0.001 | | | 0.065 |
| Never in union | 514 (2.3) | 228 (1.8) | 285 (3.2) | | 280 (3.2) | 2 (1.5) | |
| Currently in union | 20,637 (94.2) | 12,283 (95.3) | 8,354 (92.7) | | 8,125 (92.8) | 154 (91.5) | |
| Formerly in union | 760 (3.5) | 385 (3.0) | 376 (4.2) | | 354 (4.0) | 12 (7.0) | |
| **Place of residence** | | | | <0.001 | | | 0.599 |
| Urban | 8,712 (39.8) | 3,300 (25.6) | 5,412 (60.0) | | 5,251 (60.0) | 105 (62.3) | |
| Rural | 13,199 (60.2) | 9,595 (74.4) | 3,604 (40.0) | | 3,508 (40.0) | 64 (37.7) | |
| **Region of residence** | | | | <0.001 | | | 0.200 |
| North-Central | 3,031 (13.8) | 1,504 (11.7) | 1,527 (16.9) | | 1,507 (17.2) | 18 (10.8) | |

(*Continued*)

**Table 1.** (Continued)

| Characteristics | Women's most recent live birth <5 years before the 2018 NDHS (n = 21,911) | | | p-value | Women who gave birth in a health facility (n = 9,015) | | p-value |
|---|---|---|---|---|---|---|---|
| | Total | Non-facility births | Facility births | | Came from home/non-facility location | Came from another health facility | |
| | 21,911 | 12,896 (59%) | 9,015 (41%) | | 8,759 (98.1%)[#] | 168 (1.9%) | |
| | n (column %) | n (column %) | n (column %) | | n (column %) | n (column %) | |
| North-East | 3,862 (17.6) | 2,831 (22.0) | 1,031 (11.4) | | 997 (11.4) | 20 (12.1) | |
| North-West | 7,644 (34.9) | 6,393 (49.6) | 1,251 (13.9) | | 1,221 (13.9) | 29 (17.0) | |
| South-East | 2,138 (9.8) | 410 (3.2) | 1,728 (19.2) | | 1,651 (18.9) | 26 (15.2) | |
| South-South | 2,019 (9.2) | 991 (7.7) | 1,028 (11.4) | | 995 (11.4) | 27 (15.9) | |
| South-West | 3,218 (14.7) | 766 (5.9) | 2,452 (27.2) | | 2,387 (27.3) | 49 (28.9) | |
| **Pregnancy-related factors** | | | | | | | |
| **Parity** | | | | <0.001 | | | 0.112 |
| 1 | 3,758 (17.2) | 1,780 (13.8) | 1,978 (21.9) | | 1,900 (21.7) | 52 (30.7) | |
| 2–3 | 7,283 (33.2) | 3,810 (29.5) | 3,473 (38.5) | | 3,386 (38.7) | 59 (35.3) | |
| 4–5 | 5,167 (23.6) | 3,082 (23.9) | 2,084 (23.1) | | 2,035 (23.2) | 30 (17.6) | |
| 6 or more | 5,704 (26.0) | 4,224 (32.8) | 1,480 (16.4) | | 1,438 (16.4) | 28 (16.4) | |
| **Number of antenatal care visits** | | | | <0.001 | | | 0.989 |
| None | 5,336 (24.4) | 5,033 (39.0) | 303 (3.4) | | 295 (3.4) | 5 (3.1) | |
| 1–3 | 4,119 (18.8) | 2,795 (21.7) | 1,324 (14.7) | | 1,291 (14.7) | 23 (13.9) | |
| 4–7 | 8,090 (36.9) | 4,110 (31.9) | 3,980 (44.2) | | 3,883 (44.3) | 75 (44.3) | |
| 8 or more | 4,366 (19.9) | 958 (7.4) | 3,408 (37.8) | | 3,289 (37.6) | 65 (38.6) | |
| **Location of antenatal care** | | | | <0.001 | | | 0.014 |
| None | 5,336 (24.4) | 5,033 (39.0) | 303 (3.4) | | 295 (3.4) | 5 (3.1) | |
| Home | 559 (2.5) | 500 (3.9) | 58 (0.6) | | 55 (0.6) | 3 (2.0) | |
| Government hospital | 4,955 (22.6) | 2,220 (17.2) | 2,736 (30.4) | | 2,689 (30.7) | 43 (25.5) | |
| Government health centre | 7,208 (32.9) | 4,006 (31.1) | 3,202 (35.5) | | 3,122 (35.6) | 74 (44.0) | |
| Government health post/ other public sector | 755 (3.4) | 538 (4.2) | 217 (2.4) | | 207 (2.4) | 10 (5.7) | |
| Private medical sector | 3,037 (13.9) | 543 (4.2) | 2,493 (27.7) | | 2,388 (27.3) | 33 (19.6) | |
| Other | 63 (0.3) | 57 (0.4) | 6 (0.1) | | 2 (0.1) | 0.0 (0.0) | |
| **Complication woman might have experienced** | | | | | | | |
| **Woman likely to have experienced at least one complication during labour or childbirth**[*] | | | | <0.001 | | | <0.001 |
| No | 20,324 (92.8) | 12,409 (96.2) | 7,915 (87.8) | | 7,755 (88.5) | 72 (42.9) | |
| Yes | 1,587 (7.2) | 487 (3.8) | 1,1001 (12.2) | | 1,003 (11.5) | 96 (57.1) | |

(*Continued*)

**Table 1.** (Continued)

| Characteristics | Women's most recent live birth <5 years before the 2018 NDHS (n = 21,911) | | | p-value | Women who gave birth in a health facility (n = 9,015) | | p-value |
|---|---|---|---|---|---|---|---|
| | Total | Non-facility births | Facility births | | Came from home/non-facility location | Came from another health facility | |
| | 21,911 | 12,896 (59%) | 9,015 (41%) | | 8,759 (98.1%)# | 168 (1.9%) | |
| | n (column %) | n (column %) | n (column %) | | n (column %) | n (column %) | |
| **Health service accessibility** | | | | | | | |
| **Initial facility woman accessed for childbirth care** | | | | | Remained in initial facility | Went to further facility | <0.001 |
| Government hospital | | | 2,954 (32.8) | | 2,933 (33.5) | 19 (11.1) | |
| Government health centre | | | 2,920 (32.4) | | 2,871 (32.8) | 49 (29.0) | |
| Government health post/ other public sector | | | 214 (2.4) | | 196 (2.2) | 18 (10.9) | |
| Private sector | | | 2,928 (32.5) | | 2,760 (31.5) | 82 (48.8) | |
| **Final facility of childbirth** | | | | | | | <0.001 |
| Government hospital | | | 3,014 (33.4) | | 2,933 (33.5) | 79 (46.9) | |
| Government health centre | | | 2,885 (32.0) | | 2,871 (32.8) | 15 (8.7) | |
| Government health post/ other public sector | | | 196 (2.2) | | 196 (2.2) | 0.0 (0.0) | |
| Private sector | | | 2,921 (32.4) | | 2,760 (31.5) | 74 (44.4) | |
| **Mode of Transport** | | | | <0.001 | | | <0.001 |
| Stayed home/ used non-motorised transport | 14,974 (68.6) | 12,896 (100.0) | 2,078 (23.3) | | 2,074 (23.7) | 4 (2.5) | |
| Motorised transport | 6,849 (31.4) | 0 (0.0) | 6,849 (76.7) | | 6,685 (76.3) | (97.5) | |

# Excludes 88 women missing information on where they came from to deliver in the health facility

*Woman might have experienced at least one complication during labour or childbirth, and this includes women with multiple gestations (twins), women who were delivered via Caesarean section and women who suffered an early neonatal mortality (first 7 days of life)

## Characteristics of women who use motorised transport to place of childbirth

Table 2 shows that among all women (regardless of place of childbirth), there was a strong evidence of greater use of motorised transport to the childbirth location of most recent live birth among women who were aged 30–39 years, had secondary or higher education, were Christian, were in the highest wealth quintile, lived in urban areas, lived in the South-West or South-East regions, were of lower parity, reported eight or more antenatal care (ANC) visits, attended ANC in private facilities, and those who were deemed likely to have developed at least one complication of labour or childbirth (Table 2, Column B, p<0.001 for all).

Comparing the subsample of women who gave birth in a health facility (Table 2, Column D; n = 9,015) to all women in the sample (Table 2, Column B; n = 21,911), different socio-economic patterns were observed. Among those who gave birth in a health facility, there was strong evidence of greater use of motorised transport among women with no education (86%) or with secondary or higher education (76%), who practised Islam (86%), who were in the lowest (83%) or the highest (80%) wealth quintiles, and those who lived in the North West of Nigeria (96%) (Table 2, Column D, p<0.001 for all). Among women giving birth in a health facility, there was no difference in the percentage using motorised transport between urban and rural dwellers.

In terms of pregnancy-related factors, among women giving birth in a health facility, the percentages using motorised transport were highest for women having their first birth (79%),

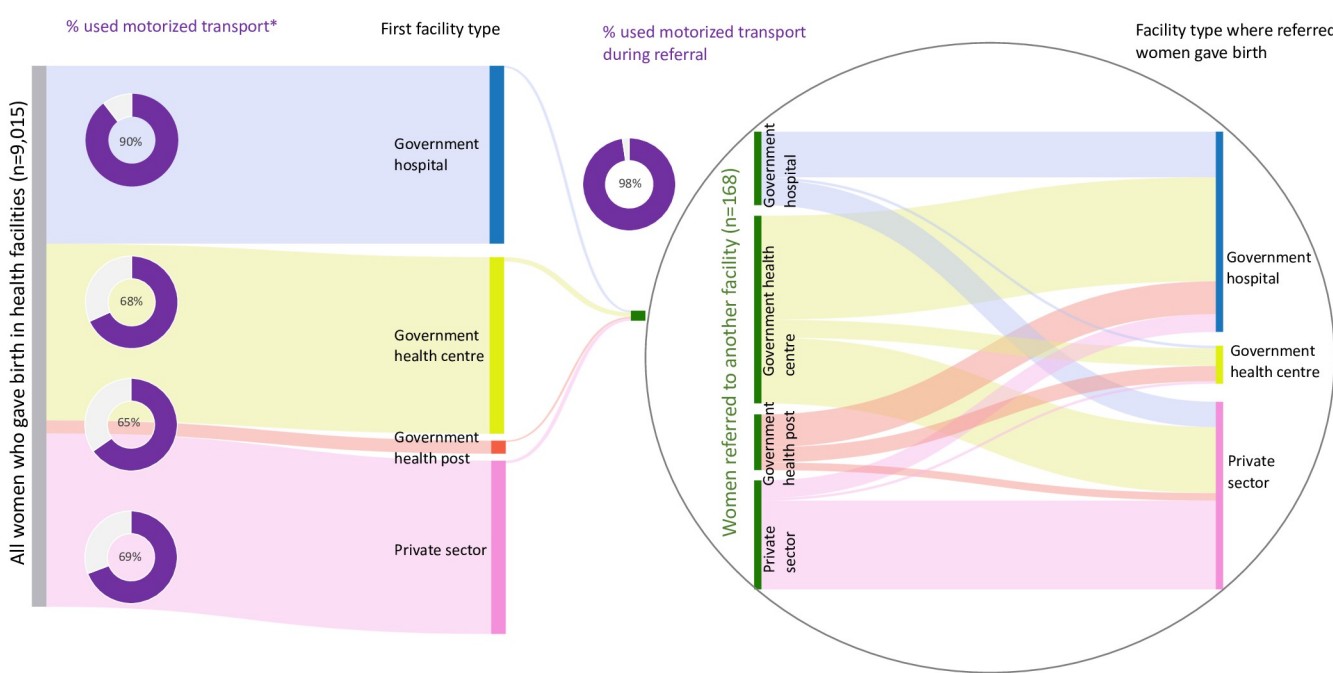

* Assuming that the mode of transport to the first facility among women who were referred out and gave birth in another facility is same as for those who delivered in the first facility

**Fig 1. Pathways to childbirth care among women whose most recent live birth was in a health facility in the five-year recall period of the 2018 Nigeria DHS (N = 9,015).**

highest among women with no antenatal care visits (80%), and highest among women who attended ANC in government hospitals (89%) (Table 2, Column D). Additionally, the use of motorised transport was common among women who were deemed likely to have experienced at least one complication of labour or childbirth (87% among women delivering in facilities vs 60% among all women in the sample). Nearly 90% of women who gave birth in government hospitals and between 68–72% of women who gave birth in private or lower-level government facilities used motorised transport to get there (Table 2, Column D).

The commonest mode of motorised transport used by women who gave birth in a health facility was the motorcycle/scooter and for women who were referred, this was a private car/ truck. There were only six ambulances reported in the sample of which three were used by women who were referred from a government health centre (See S2 Table).

## Determinants of use of motorised transport to place of childbirth

In Table 3, we present the multivariable analysis of the odds of using motorised transport to reach the place of childbirth for all women in the sample, adjusted for all other variables in the same model. Compared to women in the North-East region, the odds of using motorised transport were significantly higher for women who lived in the North-Central region (aOR = 1.68, 95% CI 1.37–2.07) and were reduced for all other regions (as low as aOR = 0.55 [95% CI 0.43–0.70] for women in the South-South region). Women who lived in rural areas were less likely to use motorised transport (aOR = 0.80, 95% CI 0.70–0.91) compared to those in urban areas. The odds of using motorised transport increased with increasing wealth quintile, education level, age, and number of ANC visits. Women were more likely to use motorised transport to get to the place of childbirth if they were deemed likely to have experienced at least one complication during labour or childbirth (aOR = 3.01, 95% CI 2.55–3.55). The odds

**Table 2. Proportion of women who use motorised transport to facility of childbirth among all women and among women who gave birth in a health facility in the 2018 NDHS.**

| Characteristics | Women's most recent live birth <5 years before the 2018 NDHS | | | Women who gave birth in a health facility | | |
|---|---|---|---|---|---|---|
| | Total sample (N = 21,823)# | Proportion of women who used motorised transport to place of childbirth N (%) (31.4%) | p-value | Total health facility births (N = 8,927)# | Proportion of women who used motorised transport to place of childbirth N (%) (76.7%) | p-value |
| | Column A | Column B | | Column C | Column D | |
| **Socio-demographic factors** | | | | | | |
| **Mother's age at birth** | | | <0.001 | | | 0.087 |
| Less than 20 | 2,665 | 630 (23.6) | | 780 | 630 (80.7) | |
| 20–29 | 10,753 | 3,396 (31.6) | | 4,424 | 3,396 (76.8) | |
| 30–39 | 7,176 | 2,493 (34.7) | | 3,291 | 2,493 (75.7) | |
| 40–49 | 1,229 | 331 (27.0) | | 432 | 331 (76.7) | |
| **Highest education attained** | | | <0.001 | | | <0.001 |
| No education | 9,731 | 1,255 (12.9) | | 1,468 | 1,255 (85.5) | |
| Primary education | 3,281 | 984 (30.0) | | 1,356 | 984 (72.6) | |
| Secondary or higher | 8,811 | 4,610 (52.3) | | 6,104 | 4,610 (75.5) | |
| **Religion** | | | <0.001 | | | <0.001 |
| Christian | 8,279 | 3,853 (46.5) | | 5,421 | 3,853 (71.1) | |
| Islam | 13,427 | 2,976 (22.2) | | 3,472 | 2,976 (85.7) | |
| Traditional/Other | 117 | 20 (16.9) | | 34 | 20 (58.6) | |
| **Wealth index** | | | <0.001 | | | <0.001 |
| Lowest | 4,715 | 454 (9.6) | | 547 | 454 (83.0) | |
| Second | 4,844 | 836 (17.2) | | 1,103 | 836 (75.8) | |
| Middle | 4,434 | 1,356 (30.6) | | 1,842 | 1,356 (73.6) | |
| Fourth | 4,076 | 1,821 (44.7) | | 2,468 | 1,821 (73.8) | |
| Highest | 3,754 | 2,382 (63.5) | | 2,967 | 2,382 (80.3) | |
| **Marital status** | | | 0.011 | | | 0.023 |
| Never in union | 511 | 190 (37.3) | | 283 | 190 (67.4) | |
| Currently in union | 20,562 | 6,393 (31.1) | | 8,279 | 6,393 (77.2) | |
| Formerly in union | 750 | 266 (35.5) | | 365 | 266 (72.9) | |
| **Place of residence** | | | <0.001 | | | 0.959 |
| Urban | 8,657 | 4,111 (47.5) | | 5,356 | 4,111 (76.8) | |
| Rural | 13,167 | 2,738 (20.8) | | 3,571 | 2,738 (76.7) | |
| **Region of residence** | | | <0.001 | | | <0.001 |
| North-Central | 3,030 | 1,284 (42.4) | | 1,526 | 1,284 (84.2) | |
| North-East | 3,849 | 858 (22.3) | | 1,018 | 858 (84.3) | |
| North-West | 7,643 | 1,193 (15.6) | | 1,250 | 1,193 (95.5) | |
| South-East | 2,088 | 1,031 (49.4) | | 1,677 | 1,031 (61.5) | |
| South-South | 2,012 | 778 (38.7) | | 1,021 | 778 (76.2) | |
| South-West | 3,202 | 1,704 (53.2) | | 2,436 | 1,704 (70.0) | |
| **Pregnancy-related factors** | | | | | | |
| **Parity** | | | <0.001 | | | <0.001 |
| 1 | 3,732 | 1,534 (41.1) | | 1,952 | 1,534 (78.6) | |
| 2–3 | 7,255 | 2,689 (37.1) | | 3,446 | 2,689 (78.0) | |
| 4–5 | 5,147 | 1,477 (28.7) | | 2,064 | 1,477 (71.6) | |
| 6 or more | 5,689 | 1,149 (20.2) | | 1,466 | 1,149 (78.4) | |
| **Number of antenatal care visits** | | | <0.001 | | | 0.001 |
| None | 5,333 | 239 (4.5) | | 300 | 239 (79.5) | |

*(Continued)*

**Table 2.** (Continued)

| Characteristics | Women's most recent live birth <5 years before the 2018 NDHS | | | Women who gave birth in a health facility | | |
|---|---|---|---|---|---|---|
| | Total sample (N = 21,823)[#] | Proportion of women who used motorised transport to place of childbirth N (%) (31.4%) | p-value | Total health facility births (N = 8,927)[#] | Proportion of women who used motorised transport to place of childbirth N (%) (76.7%) | p-value |
| | Column A | Column B | | Column C | Column D | |
| 1–3 | 4,110 | 971 (23.6) | | 1,315 | 971 (73.9) | |
| 4–7 | 8,068 | 3,151 (39.1) | | 3,957 | 3,151 (79.6) | |
| 8 or more | 4,313 | 2,489 (57.7) | | 3,355 | 2,489 (74.2) | |
| **Location of antenatal care** | | | <0.001 | | | <0.001 |
| None | 5,333 | 239 (4.5) | | 300 | 239 (79.5) | |
| Home | 558 | 33 (6.0) | | 58 | 33 (57.6) | |
| Government hospital | 4,951 | 2,417 (48.8) | | 2,732 | 2,417 (88.5) | |
| Government health centre | 7,202 | 2,251 (31.3) | | 3,196 | 2,251 (70.4) | |
| Government health post/other public sector | 755 | 157 (20.9) | | 217 | 157 (72.6) | |
| Private medical sector | 2,965 | 1,749 (60.0) | | 2,421 | 1,749 (72.2) | |
| Other | 60 | 3 (4.4) | | 3 | 3 (100.0) | |
| **Complication woman might have experienced** | | | | | | |
| **Woman likely to have experienced at least one complication during labour or childbirth** | | | <0.001 | | | <0.001 |
| No | 20,237 | 5,893 (29.1) | | 7,828 | 5,893 (75.3) | |
| Yes | 1,587 | 956 (60.3) | | 1,100 | 956 (87.0) | |
| **Health service accessibility** | | | | | | |
| **Initial facility woman accessed for childbirth care** | | | | | | <0.001 |
| Government hospital | | | | 2,951 | 2,647 (89.7) | |
| Government health centre | | | | 2,920 | 2,006 (68.7) | |
| Government health post/other public sector | | | | 214 | 146 (68.2) | |
| Private sector | | | | 2,842 | 2,050 (72.1) | |
| **Final facility of childbirth** | | | | | | <0.001 |
| Government hospital | | | | 3,012 | 2,707 (89.9) | |
| Government health centre | | | | 2,885 | 1,972 (68.4) | |
| Government health post/other public sector | | | | 196 | 128 (65.2) | |
| Private sector | | | | 2,835 | 2,042 (72.1) | |
| **Referral** | | | | | | <0.001 |
| Came from home | | | | 8,759 | 6,685 (76.3) | |
| Came from another health facility | | | | 169 | 164 (97.5) | |

[#] Excludes 88 women missing information on mode of transport to the health facility

of using motorised transport decreased with increasing parity of women. Crude odds ratios are reported in S3 Table.

In Table 4, we restricted our multivariable analysis to only women who gave birth in a facility. After adjusting for all other variables in the same model, the adjusted odds of using motorised transport when compared to women in the North-East region, were significantly higher

**Table 3. Logistic regression model for use of motorised transport to place of childbirth as an outcome in the 2018 NDHS (N = 21,823).**

| Characteristics | Adjusted model 1* | | | Adjusted Wald test for variable |
|---|---|---|---|---|
| | aOR | 95% CI | p-value | |
| **Socio-demographic factors** | | | | |
| **Region of residence** | | | | <0.001 |
| North-Central | 1.68 | 1.37–2.07 | <0.001 | |
| North-East | Ref | | | |
| North-West | 0.62 | 0.52–0.74 | <0.001 | |
| South-East | 0.73 | 0.57–0.93 | 0.011 | |
| South-South | 0.55 | 0.43–0.70 | <0.001 | |
| South-West | 0.72 | 0.56–0.93 | 0.010 | |
| **Religion** | | | | <0.001 |
| Christian | Ref | | | |
| Islam | 0.85 | 0.72–1.00 | 0.057 | |
| Traditional/Other | 0.68 | 0.38–1.19 | 0.174 | |
| **Wealth index** | | | | <0.001 |
| Lowest | Ref | | | |
| Second | 1.38 | 1.17–1.62 | <0.001 | |
| Middle | 1.88 | 1.57–2.26 | <0.001 | |
| Fourth | 2.46 | 2.02–3.00 | <0.001 | |
| Highest | 3.78 | 3.02–4.74 | <0.001 | |
| **Highest education attained** | | | | <0.001 |
| No education | Ref | | | |
| Primary education | 1.36 | 1.19–1.56 | <0.001 | |
| Secondary or higher | 1.88 | 1.63–2.16 | <0.001 | |
| **Place of residence** | | | | <0.001 |
| Urban | Ref | | | |
| Rural | 0.80 | 0.70–0.91 | 0.001 | |
| **Mother's age at birth** | | | | 0.048 |
| Less than 20 | 0.90 | 0.77–1.03 | 0.119 | |
| 20–29 | Ref | | | |
| 30–39 | 1.27 | 1.15–1.41 | <0.001 | |
| 40–49 | 1.39 | 1.13–1.70 | 0.002 | |
| **Pregnancy-related factors** | | | | |
| **Parity** | | | | <0.001 |
| 1 | Ref | | | |
| 2–3 | 0.76 | 0.67–0.87 | <0.001 | |
| 4–5 | 0.58 | 0.49–0.68 | <0.001 | |
| 6 or more | 0.55 | 0.46–0.66 | <0.001 | |
| **Number of antenatal care visits** | | | | <0.001 |
| None | Ref | | | |
| 1–3 | 4.53 | 3.78–5.43 | <0.001 | |
| 4–7 | 7.81 | 6.60–9.25 | <0.001 | |
| 8 or more | 9.76 | 8.06–11.83 | <0.001 | |
| **Complication woman might have experienced** | | | | |
| **Woman likely to have experienced at least one complication during labour or childbirth** | | | | <0.001 |
| No | Ref | | | |

*(Continued)*

**Table 3.** (Continued)

| Characteristics | Adjusted model 1* | | | Adjusted Wald test for variable |
|---|---|---|---|---|
| | aOR | 95% CI | p-value | |
| Yes | 3.01 | 2.55–3.55 | <0.001 | |

Note: aOR = Adjusted Odds Ratio; CI = Confidence Interval; Ref = Reference Category; Adjusted Wald Test: This is used in STATA to test the goodness-of-fit of a model after adding an additional variable to the model in the analysis of survey data.

*: Any variable within the model is adjusted for by all other variables within the same model

for women who lived in the North-West region (aOR = 2.61, 95% CI 1.60–4.26) and reduced for women in the southern regions (as low as aOR = 0.31 [95% CI 0.19–0.50] for women in the South-East). Women who were referred were more likely to use motorised transport (aOR = 8.87, 95% CI 1.90–41.40), as were women who were deemed likely to have experienced at least one complication of labour or childbirth (aOR = 1.86, 95% CI 1.44–2.40). There was no difference in the odds of using motorised transport comparing women in the lowest and highest wealth quintiles and comparing those with no education and those with secondary or higher. In the same adjusted model, rural women had marginally lower odds of motorised transport use (aOR = 0.86, 95% CI 0.72–1.04) compared to urban women. Women who gave birth in lower-level facilities were less likely to have used motorised transport to their place of childbirth compared to those who gave birth in government hospitals. Crude odds ratios are reported in S4 Table.

Due to the small sample size of women who were referred, we could not perform a bivariate analysis and multivariable logistic regression analysis for referral.

## Discussion

### Summary of findings

Our study showed that one-third of all women in Nigeria used some form of motorised transport to reach a health facility for their most recent live birth. Women in Nigeria were more likely to give birth in a health facility and to use motorised transport to their place of birth if they lived in the North Central region, or if they were more educated, from wealthier households, or lived in urban areas. However, when we examined motorised transport use among only women who gave birth in health facilities, different patterns of use and potential need became apparent. Roughly three-quarters of these women used motorised transport to get to health facilities for childbirth. There was no difference in the adjusted odds of using motorised between the highest and lowest wealth quintile and educational levels. Women who lived in the northern regions, women who were referred between facilities and those who experienced at least one complication of labour or childbirth, were more likely to use motorised transport to health facilities for childbirth.

As per our findings (and from the NDHS), only 41% of women give birth in health facilities in Nigeria. This is strongly patterned by region. Wong and colleagues have shown that the predicted probability of hospital birth in Nigeria was low for those with either a long travel time or from poorer households [24]. Given the high proportion of home births and the low coverage of facility births in Nigeria, the use of motorised transport to the place of childbirth follows nearly the same socio-economic patterns as health facility deliveries. It is only by examining the sub-population of women who delivered in health facilities (unadjusted model) that we find the proportions of women using motorised transport were equal or greater among women in rural areas (compared to urban), with no education (compared to secondary or

**Table 4. Logistic regression model for use of motorised transport to place of childbirth as an outcome for all women whose most recent birth was in a health facility in the 2018 NDHS (N = 8,927).**

| Characteristics | Adjusted model 2* | | | Adjusted Wald test for variable |
|---|---|---|---|---|
| | aOR | 95% CI | p-value | |
| **Socio-demographic factors** | | | | |
| **Region of residence** | | | | <0.001 |
| North-Central | 0.89 | 0.56–1.39 | 0.597 | |
| North-East | Ref | | | |
| North-West | 2.61 | 1.60–4.26 | <0.001 | |
| South-East | 0.31 | 0.19–0.50 | <0.001 | |
| South-South | 0.50 | 0.30–0.84 | 0.008 | |
| South-West | 0.30 | 0.19–0.49 | <0.001 | |
| **Religion** | | | | 0.016 |
| Christian | Ref | | | |
| Islam | 1.42 | 1.20–1.83 | 0.008 | |
| Traditional/Other | 0.61 | 0.25–1.48 | 0.271 | |
| **Wealth index** | | | | <0.001 |
| Lowest | Ref | | | |
| Second | 0.71 | 0.50–0.99 | 0.041 | |
| Middle | 0.73 | 0.51–1.05 | 0.089 | |
| Fourth | 0.82 | 0.57–1.18 | 0.297 | |
| Highest | 1.18 | 0.79–1.76 | 0.426 | |
| **Highest education attained** | | | | 0.135 |
| No education | Ref | | | |
| Primary education | 0.76 | 0.60–0.98 | 0.031 | |
| Secondary or higher | 0.82 | 0.63–1.07 | 0.144 | |
| **Place of residence** | | | | 0.022 |
| Urban | Ref | | | |
| Rural | 0.86 | 0.72–1.04 | 0.124 | |
| **Mother's age at birth** | | | | 0.945 |
| Less than 20 | 0.95 | 0.74–1.21 | 0.662 | |
| 20–29 | Ref | | | |
| 30–39 | 1.12 | 0.96–1.31 | 0.165 | |
| 40–49 | 1.17 | 0.87–1.58 | 0.290 | |
| **Pregnancy-related factors** | | | | |
| **Parity** | | | | <0.001 |
| 1 | Ref | | | |
| 2–3 | 1.02 | 0.83–1.26 | 0.819 | |
| 4–5 | 0.67 | 0.53–0.85 | 0.001 | |
| 6 or more | 0.71 | 0.53–0.96 | 0.025 | |
| **Number of antenatal care visits** | | | | 0.002 |
| None | Ref | | | |
| 1–3 | 0.66 | 0.45–0.98 | 0.049 | |
| 4–7 | 1.08 | 0.77–1.52 | 0.652 | |
| 8 or more | 1.16 | 0.81–1.66 | 0.425 | |
| **Health service accessibility** | | | | |
| **Final facility of childbirth** | | | | <0.001 |
| Government hospital | Ref | | | |
| Government health centre | 0.35 | 0.28–0.44 | <0.001 | |

*(Continued)*

**Table 4.** (*Continued*)

| Characteristics | Adjusted model 2* | | | Adjusted Wald test for variable |
|---|---|---|---|---|
| | aOR | 95% CI | p-value | |
| Government health post/other public sector | 0.23 | 0.14–0.37 | <0.001 | |
| Private sector | 0.44 | 0.35–0.55 | <0.001 | |
| **Referral** | | | | 0.002 |
| Came from home | Ref | | | |
| Came from another health facility | 8.87 | 1.90–41.40 | 0.006 | |
| **Complication woman might have experienced** | | | | |
| **Woman likely to have experienced at least one complication during labour or childbirth** | | | | <0.001 |
| No | Ref | | | |
| Yes | 1.86 | 1.44–2.40 | <0.001 | |

Note: aOR = Adjusted Odds Ratio; CI = Confidence Interval; Ref = Reference Category; Adjusted Wald Test: This is used in STATA to test the goodness-of-fit of a model after adding an additional variable to the model in the analysis of survey data

*: Any variable within the model is adjusted for by all other variables within the same model

higher) and in the lowest wealth quintile (compared to the highest). For more marginalised women, particularly in communities where facility childbirth is uncommon, women who gave birth in a health facility will likely represent either those who intended to deliver at home but then moved to a health facility because they developed a complication of pregnancy, or women who were considered high-risk and were advised to give birth in a health facility. Given their precarious conditions, such women were likely to use motorised transport to health facilities for childbirth, either because they needed to travel long distances to reach a hospital (or facility capable of managing complications) or because they were experiencing a time-sensitive emergency or could not walk. Our study showed that women who delivered in a health facility and used motorised transport to get there had nine times the odds of being referred from one facility to another (compared to those who came from home) and three times the odds of reporting at least one possible complication during labour or childbirth (multiple gestations, caesarean section, or early neonatal death) compared to women who delivered at home.

To put this finding in context, a study by Sacks and colleagues in Uganda and Zambia explored factors influencing mode of transport for obstetric care, using exit surveys of women who had a recent facility childbirth. In Uganda, women in the wealthiest quintile were more likely to travel by car or truck whereas their poorer counterparts were more likely to travel by ambulance [25]. This echoes our findings in the sense that poorer and uneducated women were more likely to seek care in a facility after developing complications of childbirth. Additionally, women in the wealthiest wealth quintile mostly used private cars or trucks to reach health facilities for childbirth. Marginalised women experiencing a complication may be more likely to need to travel to health facilities by ambulance as compared to wealthier and more educated women who were more likely not to be travelling in an emergency. The data from Uganda showed a higher prevalence of motorised transport to reach a facility for childbirth (91% compared to 76% in our study), but the commonest mode of transport by women who delivered in a health facility in Uganda and in our study in Nigeria was motorcycle taxis [4].

## Implications of our study findings

Our findings have important implications for policy, practice, and future research, especially pertaining to the design and assessment of referral and transport systems for maternal health

services. Health systems should ensure that vulnerable and less privileged women can access intrapartum services in appropriate health facilities and in a timely manner, especially for women with high-risk pregnancies who may lack access to appropriate motorised transport for referral. Facility and skilled birth attendant deliveries are disproportionately increasing for urban and richer women in LMICs [26]. With the policy focus on increasing facility and skilled birth attendant deliveries [27, 28], efforts should be made to develop affordable transport options to enable women to utilize essential childbirth care services, especially for women with an obstetric emergency who may need referral. Nigeria is currently in the process of establishing a national ambulance service [29]. However, such services need to be delivered on a large-scale and set up closer to communities. In addition, the private sector can play a significant role in helping pregnant women travel to care using motorised transport [30]. Having nearby health facilities that are not capable of managing time-sensitive complications does not provide any benefit to women, especially if they must bear the cost of motorised transport to facilities that can meet their needs. A study in Brazil reported that even under severe financial and time constraints, low-income residents travel longer to obtain access to facilities perceived as adequate to their health needs [31].

Factors such as affordability, acceptability and adequacy of transport options influence the decision of women to give birth in a health facility and the likelihood that motorised transport is used [7, 25, 32]. This means that interventions that aim to improve use of motorised transport should be affordable to poor women especially in rural areas, acceptable to women who utilize these services and protect their dignity and be adequate to serve the unmet need for such transport services. Transport and service vouchers, conditional cash transfers, and free ambulance services for instance have been considered a viable strategy for rapidly increasing access to maternal care, especially for rural and poor women [33, 34]. Additionally, policies aimed at helping women plan for childbirth, including transportation, as part of antenatal care education and counselling will likely improve the utilisation of motorised transport to facility-based childbirth. Our study exposes several potential areas of future research focus to further our understanding of travel, transport and referral patterns of women seeking childbirth care. Further studies are needed to understand where women intend to deliver, how experiencing pregnancy or childbirth complications might influence this decision and the use of different motorised transport options. Additionally, future research should leverage the possibility of using nationally representative data such as the DHS to understand pathways to childbirth care for women with all birth outcomes and women with postpartum complications.

## Strengths and limitations

A key strength of our study is the pioneering use of a nationally representative survey to describe transport and referral pathways for women accessing health facilities for childbirth and provides an essential backdrop with which we begin to understand and possibly improve transport to care systems. However, our study has some important limitations. The NDHS asked the childbirth care-seeking questions used in this analysis only of women with a live birth. There is evidence that many women needing emergency obstetric care experience stillbirth (and some women die), but these women and the pathways and modes of transport they use, if any, are not captured in this dataset [9, 22, 35]. Our study also did not provide details on the number of stops women might have made before arriving at their final facility [36, 37].

While we developed a proxy for women likely to have experienced a complication during labour or childbirth, we cannot know whether these women experienced a complication that made them change their intended place of childbirth [38]. For example, a woman reporting that she used motorised transport to go from her home to a government hospital for childbirth

could have done so because she intended to deliver at home but then experienced complications and needed urgent, life-saving care or could have done so because she always intended to give birth in the hospital and experienced no complications–there is no way to distinguish these two very different childbirth situations in the dataset. Finally, the data were also subject to potential errors with recall and reporting on transport and referral pathways, given that women were asked these questions (recalling their last pregnancy in the five years preceding the survey) with little to no means of verification.

## Conclusion

In Nigeria and in other LMICs, little is known about how women travel to reach essential and emergency care at birth when needed. Knowing the destination does not always mean knowing the path to the destination and whether this was the intended place of childbirth. Our study provides relevant information on transport and referral pathways to childbirth care for women in Nigeria using a nationally representative survey. For women who delivered in health facilities, use of motorised transport showed differing socio-economic patterns as compared to the sample of all women, that reflected both women with greater means and access (urban, more educated, and wealthier women) and women travelling to facilities because of pregnancy complications. Given that use of motorised transport to childbirth enables women get to health facilities faster, especially in an obstetric emergency and for referral, it is key that emergency obstetric transport interventions should address the unmet need in access and utilization of life-saving services by pregnant women.

## Supporting information

**S1 Checklist. STROBE checklist.**
(DOCX)

**S1 Table. Questions on transport and referral from section 4 (pregnancy and postnatal care) of the woman's questionnaire (Source: 2018 NDHS).**
(DOCX)

**S2 Table. Proportions of motorised transport options used by women who gave birth in a health facility in the 2018 NDHS.**
(DOCX)

**S3 Table. Logistic regression model for use of motorised transport to place of childbirth as an outcome in the 2018 NDHS (N = 21,823).**
(DOCX)

**S4 Table. Logistic regression model for use of motorised transport to place of childbirth as an outcome for all women whose most recent birth was in a health facility in the 2018 NDHS (N = 8,927).**
(DOCX)

**S5 Table. Proportion of women who used motorised transport to final facility of childbirth among women who were referred (N = 168) in the 2018 NDHS.**
(DOCX)

**S6 Table. Breakdown of numbers for use of motorised transport represented in the Sankey diagram (Fig 1; n = 168).**
(DOCX)

**S1 Text. Inclusivity in global research.**
(DOCX)

**S1 Data.**
(DO)

## Acknowledgments

The authors would like to thank the DHS Programme and ICF for granting us permission to use the 2018 DHS data for this analysis. We would also like to thank the Federal Ministry of Health of Nigeria for working with some the co-authors to include the questions on obstetric referral and transport to the standard woman's questionnaire of the 2018 DHS.

## Author Contributions

**Conceptualization:** Cephas Ke-on Avoka, Lenka Beňová, Emma Radovich, Oona M. R. Campbell.

**Data curation:** Cephas Ke-on Avoka, Lenka Beňová, Emma Radovich, Oona M. R. Campbell.

**Formal analysis:** Cephas Ke-on Avoka, Aduragbemi Banke-Thomas, Lenka Beňová, Emma Radovich, Oona M. R. Campbell.

**Funding acquisition:** Lenka Beňová, Oona M. R. Campbell.

**Project administration:** Cephas Ke-on Avoka.

**Supervision:** Aduragbemi Banke-Thomas, Lenka Beňová, Emma Radovich, Oona M. R. Campbell.

**Validation:** Cephas Ke-on Avoka, Aduragbemi Banke-Thomas, Lenka Beňová, Emma Radovich, Oona M. R. Campbell.

**Visualization:** Cephas Ke-on Avoka, Lenka Beňová, Oona M. R. Campbell.

**Writing – original draft:** Cephas Ke-on Avoka, Aduragbemi Banke-Thomas, Emma Radovich, Oona M. R. Campbell.

**Writing – review & editing:** Cephas Ke-on Avoka, Aduragbemi Banke-Thomas, Lenka Beňová, Emma Radovich, Oona M. R. Campbell.

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
