## [Decision Letter · Decision Letter 0]

23 Jun 2022

PGPH-D-22-00680

Use of motorised transport and pathways to childbirth care in health facilities: Evidence from the 2018 Nigeria Demographic and Health Survey

Dear Dr. Avoka,

Thank you for submitting your manuscript to PLOS Global Public Health. After careful consideration, we feel that it has merit but does not fully meet PLOS Global Public Health’s publication criteria as it currently stands. Therefore, we invite you to submit a revised version of the manuscript that addresses the points raised during the review process.

EDITOR: Please insert comments here and delete this placeholder text when finished. Be sure to:

Thank you for submitting your manuscript to PGPH for publication. Three independent reviewers have reviewed your manuscript. Although they felt it has merit, they raised substantial issues that will need your attention. There is lack of clarity in many places. There is also a need to take a second look at the data analysis, especially the issue regarding the suitability of analysing motorised transport use in non-facility births. It is important to revisit the limitations of the study. The discussion section needs to be situated within the study findings, and not opinions that are not established by study.

-Lines 137-138: How did you arrive at this decision

-Why did you present both unadjusted and adjusted ORs, when the interpretation is based on the adjusted ORs. It makes the tables unnecessarily busy. I will suggest you take out the unadjusted ORs.

Please ensure that your decision is justified on PLOS Global Public Health’s publication criteria and not, for example, on novelty or perceived impact.

We look forward to receiving your revised manuscript.

Kind regards,

Dickson Abanimi Amugsi, PhD

Academic Editor

Journal Requirements:

2. Please update the 'Competing Interests' statement with this "The authors have declared that no competing interests exist".

3. Please amend your detailed Financial Disclosure statement. This is published with the article. It must therefore be completed in full sentences and contain the exact wording you wish to be published.

- State the initials, alongside each funding source, of each author to receive each grant.

- State what role the funders took in the study. If the funders had no role in your study, please state: “The funders had no role in study design, data collection and analysis, decision to publish, or preparation of the manuscript.”

Additional Editor Comments (if provided):

Reviewers' comments:

Reviewer's Responses to Questions

**Comments to the Author**

1. Does this manuscript meet PLOS Global Public Health’s publication criteria? Is the manuscript technically sound, and do the data support the conclusions? The manuscript must describe methodologically and ethically rigorous research with conclusions that are appropriately drawn based on the data presented.

Reviewer #1: Partly

Reviewer #2: Yes

Reviewer #3: Yes

2. Has the statistical analysis been performed appropriately and rigorously?

Reviewer #1: No

Reviewer #2: Yes

Reviewer #3: Yes

3. Have the authors made all data underlying the findings in their manuscript fully available (please refer to the Data Availability Statement at the start of the manuscript PDF file)?

Reviewer #1: Yes

Reviewer #2: No

Reviewer #3: Yes

4. Is the manuscript presented in an intelligible fashion and written in standard English?

Reviewer #1: Yes

Reviewer #2: Yes

Reviewer #3: Yes

5. Review Comments to the Author

Reviewer #1: It is my pleasure to review this important article in the context of maternal health service utilization using some mechanism.

General comment:

This is really important and need to make understandable on what and how to use such transport mechanisms in the far health facilities. I was interested if the study includes the proportion of the motors, vehicles or ambulances availability and its utilization by mothers. If you can include the availability of motors in this study, that can help the readers to understand transport utilization in health system.

Comment 1: abstract –

You put the recommendations – Obstetric transport interventions should be affordable, adequate and bridge the equity gap in utilisation of these services; but I couldn’t get any result in that aspect.

Comment 2: results – Table 2

Analyzing motorized transport use in non-facility births is not good; better to remove this part. The first thing – they may want to use and not accessible – no information about that; the second thing – it make confusion for the readers – different results in one article.

Comment 3: limitation part

It is understandable that when you use secondary data you can draw multiple limitations, so better to minimize the limitations to increase the study creditability.

Comment 4: your study objective was specifically on motorized transport use in non-facility births, but you assessed and presented several issues including place of birth, referring places and its determinants. So, it is better to consistent across the document including study objective, results, discussion, conclusion, and recommendations.

Reviewer #2: Review for PLOS Global Public Health PGPH-D-22-00680

Lynn M. Crosby, Ph.D.

“Use of motorized transport and pathways to childbirth care in health facilities: Evidence from the 2018 Nigeria Demographic and Health Survey”

By Avoka CK, Banke-Thomas A, et al.

Summary

This is a study of the type of transport used by women who delivered a live baby within the past 5 years in Nigeria and is based on recall by the parent at the time of questionnaire. It uses descriptive and regression statistics and presents results in the form of odds ratios, which is to say, likelihood compared to a baseline value. The questionnaire was a nationally administered and sponsored instrument with thousands of responses. The study results potentially can be used to help shape future social policies that may improve maternal or neonatal health outcomes.

Issues/challenges/shortcomings:

1. The study is based on a questionnaire employing personal recall from the previous 5 years. The subjects filling out the questionnaire could mis-remember, mis-understand the questions, or be in a hurry and fill out the questionnaire hastily, either misstating or omitting information. Thus, the personal recall questionnaire is notoriously prone to error.

In addition, five years is too long to have accurate recall and there was no method of substantiating or corroborating the recalled answers. I would have considered the preceding 3 years as reasonably reliable and the preceding 1 year as highly reliable.

Finally, the period over five years could include substantial changes in societal capacities and behaviors that affect the results.

2. It is unclear how rates of literacy affected the results of this study but seems clear that one cannot fill out a questionnaire without reading/writing skills, every country has a finite illiteracy, and yet there was no uncertainty added for this probability or any discussion of its likelihood. For those who were illiterate, was assistance filling the form provided or permitted? Or were all forms filled out by a second person asking the questions?

3. For those 88 questionnaires with missing information, they should not have been included. Instead, some information was inferred for the missing data, but they were still included in the study. Inferring results when not known is inappropriate. Although the results might not have tipped the balance to a different outcome, it seems there were enough responses with complete information that the 88 responses were not required to reach statistically reliable conclusions.

4. The study is a cross-sectional study which is not the best type of study as (for one thing) there is no control group included. The study design makes drawing definitive conclusions problematic.

5. Some of the results could easily be attributed to reverse-causation. That should be pointed out.

6. There were some results that seemed trivial or obvious. For instance, among women transferred from another health facility 98% used motorized transport. But it would seem obvious that transfer from one facility to another is a result of an obstetrical complication, and it would not be possible to use a non-motorized transport during a health emergency. How useful is this result? How useful is the study if it only tells one what must be so? If the statistics confirm for us women living in more affluent areas, have higher education, or come from the most affluent quintile are more likely to use motorized transport, is that a surprise? Is it valuable information? If so, how may it be used? If it is not ‘new’ information, the reader’s interest is not captured and the journal impact is lessened.

7. It would have been useful if the questionnaire had asked about motivations for using or not using motorized transport to obtain delivery care. As the authors pointed out, the facts alone do not reveal whether women intended to use motorized transport or not and how their plans for transportation worked or did not work at the time they gave birth. For instance, the following statement was made:

“Understanding travel to reach health facilities and factors that influence transport and referral mechanisms is a critical step needed to develop an effective obstetric transport and

referral system [12].”

Does this study reveal the factors that influence the transport that was used? It is unclear.

8. Because the data were expressed as odds ratios, they were expressed as relative to some data categorical result, arbitrarily set to 1. Since the choice of reference is arbitrary, this would necessarily affect the results. Should be pointed out and discussed.

9. At the end of reading the manuscript, my question was how can these data be used? The statement above was that understanding travel to reach health facilities is a critical step to develop effective obstetric transport. How and where do these results fit into fulfilling that goal?

The audience is missing critical statements that would connect the dots between the important research questions and the stated goals of better obstetrical transport pathways. Not sure that the data collected, and their analysis was able to answer the questions.

10. Restriction to live births information, and from living mothers who gave birth, is going to skew the data because the worst outcomes, the unthinkably bad outcomes, are those where transport affects the viability of mother or offspring, yet these were not collected. I feel this does skew the data toward the less severe outcomes and muddles the truth. The authors touch on this in their last section of the discussion.

11. The statement was made that “Obstetric transport interventions should be affordable, adequate and bridge the equity gap in utilisation of these services.” However, this study did not measure the affordability or adequacy in utilization of motorized services. Affordability would be measured by responses to questions about the cost of transport, but here there were none. Adequacy would be measured by responses to questions about the plans for transport before birth and the availability of choices, together with the actual option used and justification for using that option, or alternatively would include information about the total number of motorized vehicles available for hire or of public or private transport capacity of the system at the time of birth. But these questions were not part of the study. Therefore, the statement has nothing to do with the study results. It is an opinion or judgment and is not a supported conclusion from these data.

12. Although the STROBE checklist was provided, not all categories contained a response.

13. The Discussion section contains quite a bit of opinion that seems to be designed to support policy decisions. I don’t really have a problem with any of the statements, they seem well-intentioned. However, the authors should be cautioned that the Discussion should be directly tied to the data presented in the manuscript and not depart far from the factual conclusions that can be directly reached using these data. That is overreaching.

Strengths:

Uses a national survey which one would assume to be carefully organized and conducted. It is a well-written manuscript. The statistical methods employed appear valid and adequate. Organization of the results and their presentation was good.

Certain of the conclusions and observations are very salient, e.g., “Our study showed that women who delivered in a health facility and used motorised transport to get there had nine times the odds of being referred from one facility to another (compared to those who came from home) and three times the odds of reporting at least one possible complication during labour or childbirth (multiple gestation, caesarean section or an early neonatal death) compared to women who delivered at home.”

Data Analysis:

Descriptive, crude and adjusted logistic regression analyses were conducted to assess the determinants of use of motorized transport.

Specific suggestions for improvement (e.g., line-by-line):

For Table 1 (Sociodemographic, obstetric factors and transport pathways….) it is not clear to me which comparison the p-value is referring to, as it is listed on the first line in each variable. For instance, ‘mother’s age at birth’ has a p-value for the ‘less than 20’ line entry of <0.001 just to the right of the block titled “women’s most recent live birth <5 years before the 2018 NDHS (n=21,911), so I think this value applies to the comparison of non-facility births vs. facility births among <20 years old, but what about the groups 20-29, 30-39, 40-49? There is no p-value listed beside these lines, what does that mean? Does it mean the comparison wasn’t made, wasn’t significant, or some other conclusion? The remainder of the data table is laid out the same way, with only a p-value in the first line of the variable.

The same question re Table 2. There is no explanation in the text for the way the p-value is presented, and it is not intuitive. On page 14, line 180-181, the statement is made that “(Table 2, column B, p<0.001 for all)”. Then, if the p-value is the same for each comparison in the group and for all groups, it should be stated in the table somewhere. Otherwise, one cannot tell even if all the comparisons were run.

And if it is true the p-value is the same for every comparison within the group, for all groups (all are presented the same way), it seems a little strange. Statistical comparisons are rarely that neat and tidy.

Table 3. For each categorical variable, one choice was chosen as the index and assigned a value of “1”. Was there a basis for this choice? What was it? As mentioned above, the presentation of the results is biased by the choice of index.

Table 3 – The column “Adjusted Wald test for variable” needs further explanation in the footnote or within the text. It would be easier for the reader if this is included in the footnote to the table, to save having to hunt for it in the text. What is the value of performing this adjustment and what happens to the result when it is done? It this adjustment critical or of value? Based on a cursory comparison of the unadjusted and adjusted Wald test columns, there is not a difference in the result, and inclusion does not therefore seem to be of value. It just adds more columns to the table without providing further information. You could add a note in the text or footnote that the Wald test was performed and yielded the same results.

Same comments for Table 4 as for Table 3.

S4 Table – n= 168 the total number of observations adds to 165.

S3 Table – please check the math, not all categories sum to 168.

Decision/Recommendation:

The data and conclusions can stand alone, however, I suggest that the authors may want to collect further information that would make the manuscript more interesting and informative.

Editor can make the final decision once adequate responses to the concerns listed above are provided. I feel that the paper is somewhat light on importance and impact, but not without value. It is just so-so.

Some of the design features of the study make it less powerful (personal recall, cross-sectional).

Reviewer #3: The article sufficiently provides information and evidences on transport and referral pathways to obstetric and childbirth care for women in Nigeria including recommendation for affordable, acceptable and adequate transport intervention during such emergency which can also be relevant to the developing countries.

The further comments and feedbacks includes:

The formatting of the article needs to be improved as prescribed by the journal guidelines while language and punctuation improvements are required in the introduction and findings.

The line 73/74 would be better written with punctuation as”, Despite the recognition of critical research needs; to better understand how women access essential care at child birth while referral and transport systems remain poorly documented and researched”.

In line 81 author mentioned the adaptation of the standard Women’s Questionnaire for country priorities. Please mention for what purpose was it adapted and what is the relevance of this particular evidence in current study? Please make the clear mention.

The methodology has been very much rigorously written while I would like the author to clearly mention the rationale/logic behind recoding the missing values as such.

Punctuation missing in the line no 168 could better be rephrased for clear reading.

The presentation of the findings is very fine and I hope it meets up the journal requirements. The findings could be discussed with more other relevant articles in context of developing and developed countries as well. I hope authors could find relevant publications as well regarding the pathway of delivery.

6. PLOS authors have the option to publish the peer review history of their article (what does this mean?). If published, this will include your full peer review and any attached files.

**Do you want your identity to be public for this peer review?** For information about this choice, including consent withdrawal, please see our Privacy Policy.

Reviewer #1: No

Reviewer #2: **Yes: **Lynn M. Crosby

Reviewer #3: **Yes: **Rabindra Bhandari

---

## [Decision Letter · Decision Letter 1]

22 Aug 2022

Use of motorised transport and pathways to childbirth care in health facilities: Evidence from the 2018 Nigeria Demographic and Health Survey

PGPH-D-22-00680R1

Dear Dr Avoka,

We are pleased to inform you that your manuscript 'Use of motorised transport and pathways to childbirth care in health facilities: Evidence from the 2018 Nigeria Demographic and Health Survey' has been provisionally accepted for publication in PLOS Global Public Health.

Best regards,

Dickson Abanimi Amugsi, PhD

Academic Editor

Reviewer Comments (if any, and for reference):

Reviewer's Responses to Questions

**Comments to the Author**

1. If the authors have adequately addressed your comments raised in a previous round of review and you feel that this manuscript is now acceptable for publication, you may indicate that here to bypass the “Comments to the Author” section, enter your conflict of interest statement in the “Confidential to Editor” section, and submit your "Accept" recommendation.

Reviewer #1: All comments have been addressed

Reviewer #3: All comments have been addressed

2. Does this manuscript meet PLOS Global Public Health’s publication criteria? Is the manuscript technically sound, and do the data support the conclusions? The manuscript must describe methodologically and ethically rigorous research with conclusions that are appropriately drawn based on the data presented.

Reviewer #1: Yes

Reviewer #3: Yes

3. Has the statistical analysis been performed appropriately and rigorously?

Reviewer #1: Yes

Reviewer #3: Yes

4. Have the authors made all data underlying the findings in their manuscript fully available (please refer to the Data Availability Statement at the start of the manuscript PDF file)?

Reviewer #1: Yes

Reviewer #3: Yes

5. Is the manuscript presented in an intelligible fashion and written in standard English?

Reviewer #1: Yes

Reviewer #3: Yes

6. Review Comments to the Author

Reviewer #1: Thank you for giving the chance to review again! The manuscript has been well improved and adequate enough to publish!

Reviewer #3: Thank you for consideration and addressing all the comments adequately. I hope journal would require additional proof reads before publication.

7. PLOS authors have the option to publish the peer review history of their article (what does this mean?). If published, this will include your full peer review and any attached files.

**Do you want your identity to be public for this peer review?** For information about this choice, including consent withdrawal, please see our Privacy Policy.

Reviewer #1: No

Reviewer #3: **Yes: **Rabindra Bhandari
